# Successful Catheter Ablation of the “R on T” Ventricular Fibrillation

**DOI:** 10.3390/ijerph18189587

**Published:** 2021-09-11

**Authors:** Zofia Lasocka, Alicja Dąbrowska-Kugacka, Ewa Lewicka, Aleksandra Liżewska-Springer, Tomasz Królak

**Affiliations:** Department of Cardiology and Electrotherapy, Medical University of Gdańsk, 80-211 Gdansk, Poland; zofia.lasocka@gumed.edu.pl (Z.L.); ewa.lewicka@gumed.edu.pl (E.L.); aleksandra.lizewska-springer@gumed.edu.pl (A.L.-S.); tomasz.krolak@gumed.edu.pl (T.K.)

**Keywords:** ventricular fibrillation, R on T phenomenon, catheter ablation, mapping

## Abstract

In patients with idiopathic ventricular fibrillation (VF), recurrent implantable cardioverter-defibrillator (ICD) shocks might increase mortality risk and reduce patients’ quality of life. Catheter ablation of triggering ectopic beats is considered to be an effective method. We present a patient with recurrent VF, caused by the “R on T” premature ventricular complexes. In the presented case radiofrequency catheter ablation efficiently eliminated arrhythmia trigger, which was possible to detect thanks to the intracardiac electrocardiograms (ECG’s) stored in the ICD.

## 1. Introduction

Idiopathic ventricular fibrillation (VF) occurs in healthy individuals without structural heart disease. Although most sudden cardiac deaths (SCDs) are associated with identifiable causes, idiopathic VF accounts for 5% to 10% of out-of-hospital cardiac arrest [1]. In such patients, the gold standard treatment for either primary or secondary prevention of VF is the insertion of an implantable cardioverter-defibrillator (ICD). The goal is however to determine the mechanism of spontaneous arrhythmia. Intracardiac electrocardiograms (IECGs) stored in ICD provide information about cardiac rhythm preceding arrhythmic events and may be helpful to reveal their mechanism [2].

Mapping during VF reveals few sources triggering the arrhythmia. In patients with idiopathic VF, premature ventricular complexes (PVCs) triggers usually arise either from right ventricular outflow tract region or from the Purkinje network [3]. The Purkinje network fibers originate from the bundle branches, whose function is to distribute the depolarization wavefront to the left and right ventricles, allowing for their simultaneous activation. However, Purkinje cells have shown to have abnormal automaticity and triggered activity, what may play a significant role in the initiation of VF in patients with both structural heart disease and normal hearts [4,5,6].

As recurrent ICD shocks might increase mortality risk and reduce patients’ quality of life, anti-arrhythmic drugs are recommended [7]. According to the Optimal Pharmacological Therapy in Cardioverter Defibrillator Patients (OPTIC) trial, which included patients with secondary prevention ICD indication, amiodarone compared to b-blocker reduced the risk of both appropriate and inappropriate shocks [8]. However, if pharmacological treatment is not successful or the patient experiences drug-related side effects, radiofrequency (RF) ablation can be considered if VF is triggered by PVCs [3,4,5].

## 2. Case Report

We present the history of a 62-year-old male with recurrent VF triggered by the “R on T” ventricular premature beats. In 2006, he underwent cardiac arrest and was diagnosed with idiopathic VF, as no apparent structural cardiac disease or other cause of the arrhythmia was found. Twelve-lead surface electrocardiogram (ECG) showed sinus rhythm, with normal QRS complexes and repolarization, while 24-hour ECG monitoring revealed multiple monomorphic PVCs (<1000) and no other abnormalities. In the echocardiographic examination good left ventricular ejection fraction (LVEF) of 70%, without any segmental hypo/akinesis was found. A coronary angiogram revealed no narrowing in the coronary arteries. Cardiac magnetic resonance was not performed in 2006, as it was not widely available at that time. The patient was implanted with a single-chamber ICD for secondary prevention of sudden cardiac death. At hospital discharge, anti-arrhythmic treatment with amiodarone 200 mg and metoprolol 50 mg daily was prescribed.

Within the next year, amiodarone was withheld due to thyreotoxicosis. Thyroid function was normalized, metoprolol dose was increased to 175 mg daily, but VF episodes occurred about one to two times per year and were terminated by ICD shock. As presented on Figure 1, the 12-lead ECG revealed singular PVCs with morphology of right ventricular free wall origin. The initial ICD was replaced in 2016 due to battery replacement indications. Stored IECGs revealed VF episodes initiated by the “R on T” PVCs and terminated spontaneously, by anti-tachycardia pacing (ATP) or high-voltage shocks (Figure 2A,B). To reduce the number of painful ICD interventions, as pharmacological treatment was not efficient, the patient was referred for elective electrophysiological study (EPS) and RF ablation of ectopic beats. Pre-ablation 24-hour ECG monitoring showed sinus rhythm, 1629 single monomorphic PVCs with around 300 qualified as “R on T”.

Ablation catheter (Thermocool Smarttouch SF, Biosense Webster, Johnson and Johnson Medical, Ltd., Irvine, CA, USA) was introduced to the right ventricle through a femoral approach under fluoroscopic guidance. Using CARTO 3 mapping system (version 7.1, Biosense Webster, Irvine, CA, USA) three types of right ventricular endocardial maps were created: the bipolar map during sinus rhythm, the correlation map of pace-mapping (using PaSo module), the activation map during clinical PVCs. The bipolar map revealed no abnormalities. The earliest endocardial activation site and maximal pace-mapping correlation (93% according to PaSo module) was found in the antero-lateral right ventricular free wall (Figure 2C). RF ablation current was delivered at 30 W of power-controlled mode with a temperature of 43 °C for 30–60 sec. Clinical spontaneous PVCs were eliminated, but pacing delivered from right ventricular lead of ICD occasionally induced single PVCs with slightly different morphology compared to the clinical PVCs.

In the follow-up, until June 2021 the patient remained asymptomatic and without episodes of VF or ventricular tachycardia in the ICD control. The patient’s quality of life increased significantly. 24-hours ECG monitoring revealed sinus rhythm and only eight PVCs. Metoprolol was gradually reduced to 50 mg daily.

## 3. Discussion

Our case demonstrates the efficacy of RF ablation in a patient with recurrent idiopathic VF, and ineffective antiarrhythmic treatment. IECGs from the ICD provided information about the mechanism of the arrhythmic episodes, which in the presented case was the “R on T” phenomenon. Indeed, electrocardiogram stored in the ICD devices is comparable to continuous Holter monitoring. Information, such as the day and time of the episode, the preceding heart rate, the influence of preceding premature beats and their morphology can be obtained from the IECG records analysis. In other words, it enables the physician to identify the arrhythmia trigger, underlying recurrent VF or ventricular tachycardia, and consequently, to apply the most appropriate treatment [2].

In up to 70% of patients after ICD implantation antiarrhythmic agents need to be initiated, in order to treat atrial tachyarrhythmias, terminate ventricular arrhythmias and decrease the frequency of ICD shocks. Class III antiarrhythmic drugs, such as amiodarone and sotalol, are widely considered to be effective in preventing ICD shocks [9]. However, potential cardiac and drug-related adverse effects of antiarrhythmics should be taken into consideration, as apart from ICD/drug interactions, these are the most frequent causes of drug discontinuation. In our case amiodarone was withheld due to thyreotoxicosis. The concomitant pharmacological therapy turned out insufficient to prevent recurrent ventricular tachyarrhythmias.

To reduce the number of ICD shocks, catheter ablation of the triggering PVCs was recommended. Pace-mapping allowed identification of the PVCs’ origin. According to recent reports, PVCs triggering VF arise either from the myocardium (right or left ventricular outflow tract) or from the distal Purkinje network [4]. In the study on 27 patients with recurrent episodes of idiopathic VF, PVCs originated from the Purkinje conducting system in 23 patients, while from right ventricular outflow tract only in 4 study counterparts [3]. There is a growing body of evidence that the Purkinje network, consisting of a single branch on the right and 2 larger branches on the left heart side, plays a significant role in both the initiation and maintenance of VF. It has been proved that RF ablation of arrhythmia triggers effectively prevent VF recurrence in a high-risk population [5,6]. Decreasing the incidence of ventricular arrhythmias catheter ablation may reduce defibrillation requirement and improve the patient’s quality of life.

## 4. Conclusions

In conclusion, implantation of ICD is the gold standard treatment for both primary and secondary prevention of SCD. The primary goal is to identify the mechanism underlying the spontaneous arrhythmia. In the presented case, RF ablation efficiently eliminated the arrhythmia trigger, which was identified thanks to the stored intracardiac electrograms. RF ablation is considered an effective method of arrhythmia termination, in case of antiarrhythmic drugs intolerance or inappropriate ICD shocks. Although the long-term results are very encouraging, ablation of triggering PVCs for VF does not replace ICD implantation.

## Figures and Tables

**Figure 1 ijerph-18-09587-f001:**
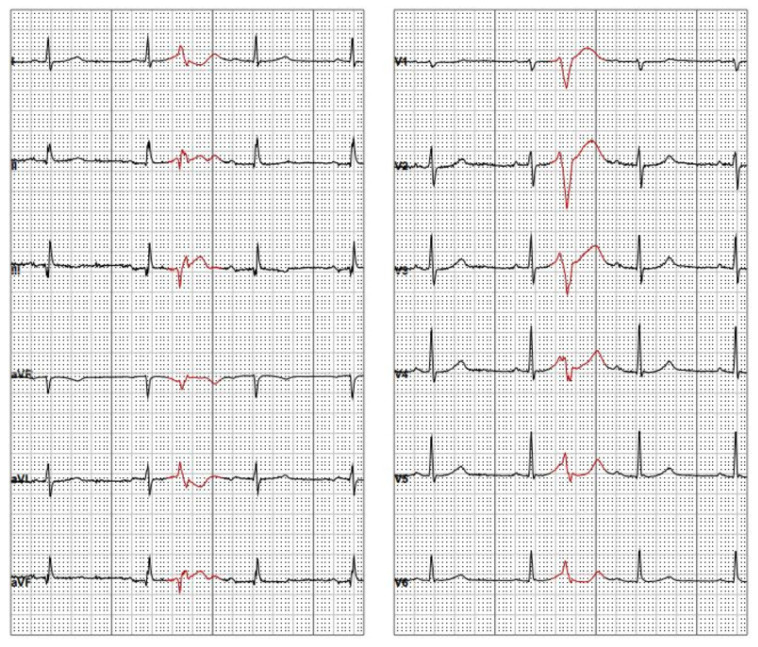
12-lead electrocardiogram (ECG) with premature ventricular complex (PVC) originating from the right ventricular free wall.

**Figure 2 ijerph-18-09587-f002:**
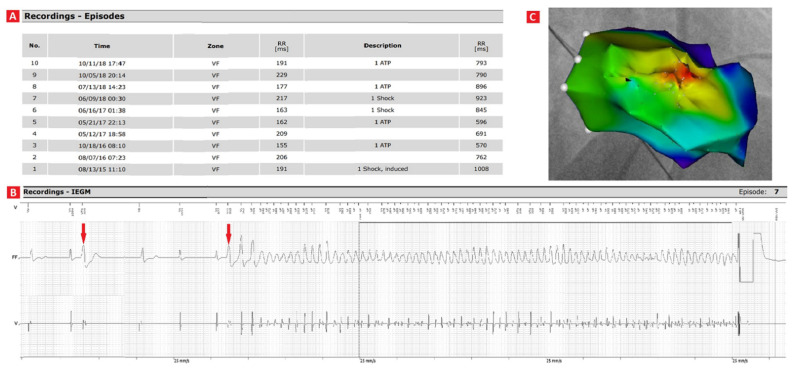
Ventricular fibrillation (VF) episodes resulting in implantable cardioverter-defibrilator (ICD) interventions, from the time of ICD reimplantation to the decision to perform radiofrequency ablation (**A**); ICD intracardiac recording showing the premature ventricular complex (PVC) triggering VF. VF initiating PVC is similar to the previously registered PVC (arrows) (**B**); Right anterior oblique 30° view of right ventricular endocardial activation map obtained during spontaneous PVCs: the earliest activation site was located in the antero-lateral right ventricular free wall (red area) (**C**). ATP—antitachycardia pacing, IEGM—intracardiac electrocardiogram, RR—consecutive R waves on the IEGM.

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
