# Peer review of "Successful Catheter Ablation of the “R on T” Ventricular Fibrillation"

_ijerph, 2021, doi:10.3390/ijerph18189587_

Round 1

Reviewer 1 Report

It would be interesting to include the 12 leads of PVC inducing VF or at least the morphology of pvc. I think an ECG figure should be added.

The authors should also explain that they were able to correlate the ECG and EGMs stored in the ICD. In this way, they could affirm that "RF ablation efficiently eliminated the arrhythmia trigger, which was identified thanks to the stored intracardiac electrograms".

Reviewer 2 Report

Thank you for providing me the opportunity to review the paper entitled "Successful catheter ablation of the “R on T” ventricular fibrillation".

General comments:

The study reported successful catheter ablation of the “R on T” ventricular fibrillation. The authors proposed that In the presented case, RF ablation efficiently eliminated the arrhythmia trigger, which was identified thanks to the stored intracardiac electrograms. RF ablation is considered an effective method of arrhythmia termination, in case of antiarrhythmic drugs intolerance or inappropriate ICD shocks. It is a well-written case report, and the findings are well presented. I have some comments.

Major:

  1. The authors described that the earliest endocardial activation site and maximal pace-mapping correlation (93% according to PaSo module) was found in the antero-lateral right ventricular free wall. How much earlier activation site did the authors ablate compared to clinical PVC? And did the authors have 12-leads ECG of clinical PVC that could localize the PVC origin?
  2. RV Purkinje VF triggers display an uniform morphology which has first been reported by Haïssaguerre et al (Circulation 106(8):962–967). who pioneered mapping and ablation of Purkinje-related idiopathic VF. In addition, Haïssaguerre et al (Nat Rev Cardiol 13(3):155–166.). provided non-invasive ECG imaging data of RV Purkinje triggers that point to the Moderator band. I just wonder if in this case, the authors did ablation near moderator band (MB)
  3. if so, As the right bundle branch and its Purkinje network run along with the MB and arborize via the MB at the free wall, it is possible to record myocardial as well as Purkinje potentials in this region during EPS. I wonder if the authors could find these kind of Purkinje potentials near MB during ablation and did these potential guided ablation?

Minor

  1. The authors described that coronary angiogram revealed no narrowing. Coronary spasm test has done in this patient? If so, please add on those.
  2. is there any chance that the patient’s thyroid function could be normalized during the follow-up?

I just wonder if the patient’s PVC trigger could be associated with thyrotoxicosis. How was the patient’s thyroid function when PVC ablation was done?
